# Pre-existing chromosomal polymorphisms in pathogenic *E. coli* potentiate the evolution of resistance to a last-resort antibiotic

**Pramod K Jangir[1]\*[†‡], Qiue Yang[2†], Liam P Shaw[1], Julio Diaz Caballero[1], Lois Ogunlana[1], Rachel Wheatley[1], Timothy Walsh[1], R Craig MacLean[1]\***

[1]Department of Zoology, University of Oxford, Oxford, United Kingdom; [2]Fujian Provincial Key Laboratory of Soil Environmental Health and Regulation, College of Resources and Environment, Fujian Agriculture and Forestry University, Fuzhou, China

**\*For correspondence:**
jangirk.pramod@gmail.com
(PKJ);
craig.maclean@zoo.ox.ac.uk
(RCraigMacL)

[†]These authors contributed equally to this work

**Present address:** [‡]Biozentrum, University of Basel, Basel, Switzerland

**Competing interest:** The authors declare that no competing interests exist.

**Abstract** Bacterial pathogens show high levels of chromosomal genetic diversity, but the influence of this diversity on the evolution of antibiotic resistance by plasmid acquisition remains unclear. Here, we address this problem in the context of colistin, a 'last line of defence' antibiotic. Using experimental evolution, we show that a plasmid carrying the MCR-1 colistin resistance gene dramatically increases the ability of *Escherichia coli* to evolve high-level colistin resistance by acquiring mutations in *lpxC*, an essential chromosomal gene involved in lipopolysaccharide biosynthesis. Crucially, *lpxC* mutations increase colistin resistance in the presence of the MCR-1 gene, but decrease the resistance of wild-type cells, revealing positive sign epistasis for antibiotic resistance between the chromosomal mutations and a mobile resistance gene. Analysis of public genomic datasets shows that *lpxC* polymorphisms are common in pathogenic *E. coli,* including those carrying MCR-1, highlighting the clinical relevance of this interaction. Importantly, *lpxC* diversity is high in pathogenic *E. coli* from regions with no history of MCR-1 acquisition, suggesting that pre-existing *lpxC* polymorphisms potentiated the evolution of high-level colistin resistance by MCR-1 acquisition. More broadly, these findings highlight the importance of standing genetic variation and plasmid/chromosomal interactions in the evolutionary dynamics of antibiotic resistance.

## Editor's evaluation

This paper combines evolution experiments with genomic analysis of environmental samples to study the evolution of colistin resistance in *Escherichia coli*. It highlights the importance of pre-existing genomic variations on the chromosomes of clinical strains in driving the evolution of antibiotic resistance. The results are relevant for clinical and non-clinical microbiologists studying antibiotic resistance to last-resort drugs like colistin. The design of the research is simple and elegant, and the genomic data analysis connects the in vitro findings to the real world.

## Introduction

Decades of research on the genetics of antibiotic resistance have shown that pathogenic bacteria evolve resistance to clinically relevant doses of antibiotics by acquiring a small number of mutations or by acquiring mobile elements carrying dedicated antibiotic resistance gene (***MacLean and San Millan, 2019***; ***Partridge et al., 2018***). An open and unresolved question in evolutionary biology and

genetics is the extent to which the other genes in bacterial genomes contribute to the evolution of resistance by modifying the rate of acquisition and effects of resistance determinants (*Lukačišinová et al., 2020*; *Vogwill et al., 2014*; *Bethke et al., 2020*; *Card et al., 2019*; *Vogwill et al., 2016*; *Bottery et al., 2017*; *MacLean et al., 2010*; *Torres Ortiz et al., 2021*).

Many of the bacterial pathogens where resistance poses a high risk to public health are opportunistic pathogens with high levels of chromosomal genetic diversity (*Baker et al., 2018*; *Didelot et al., 2012*; *Smith et al., 2000*). Although these pathogens acquire resistance determinants against a backdrop of extensive chromosomal diversity, the importance of this diversity in shaping the evolution of resistance remains poorly understood (*Torres Ortiz et al., 2021*; *Baker et al., 2018*; *Didelot et al., 2012*; *Smith et al., 2000*; *Papkou et al., 2020*). One approach to addressing this problem has been to challenge diverse collections of clinical/environmental strains with evolving resistance to a controlled antibiotic pressure using laboratory experimental evolution (*Vogwill et al., 2014*; *Bethke et al., 2020*; *Papkou et al., 2020*; *Gifford et al., 2018*). These studies have shown that strains evolve resistance at different rates, and it has been possible to identify 'potentiator' genes that accelerate the evolution of resistance by epistatically increasing the effects of canonical chromosomal resistance mutations (*Papkou et al., 2020*; *Gifford et al., 2018*). For example, strains of *Staphylococcus aureus* with high levels of expression of the *norA* efflux pump rapidly evolve resistance to fluoroquinolone antibiotics because high levels of *norA* expression increase the effect of classical fluoroquinolone resistance mutations in DNA topoisomerase (*Papkou et al., 2020*). Although these studies have focussed on resistance evolution by chromosomal mutations, most resistance in important pathogenic bacteria is caused by the acquisition of plasmids carrying mobile resistance genes (*MacLean and San Millan, 2019*; *Partridge et al., 2018*). It is equally conceivable that chromosomal polymorphisms shape the evolution of resistance by altering the acquisition and fitness effects of horizontally acquired resistance genes (*Bottery et al., 2017*).

One important challenge of using experimental evolution to identify potentiator genes is that pathogen isolates often differ from each other by thousands of single-nucleotide polymorphisms (SNPs) and variation in the presence/absence of accessory genes. Previous studies have used transcriptomic methods and genome-wide association study to identify genes that cause variation in evolvability between strains, but identifying these causal variants is challenging using this approach (*Torres Ortiz et al., 2021*; *Baker et al., 2018*; *Didelot et al., 2012*; *Smith et al., 2000*; *Papkou et al., 2020*). Given that potentiator genes interact epistatically with resistance genes, it follows that the presence of a resistance gene should also generate selection on potentiator genes under antibiotic pressure. According to this logic, comparing genomic changes between populations that carry established resistance determinants and sensitive control populations that arise during selection for elevated antibiotic resistance should provide a simple way to search for potentiator genes that interact epistatically with classical 'resistance' genes.

To test this idea, we challenged populations of *Escherichia coli* carrying an MCR-1 (mobile colistin resistance) plasmid and wild-type control populations with adapting to colistin, a last line of defence antibiotic that is used in the treatment of infections caused by multidrug-resistant pathogens (*Li et al., 2006*). *E. coli* has adapted to the widespread use of colistin in agriculture by acquiring a range of conjugative plasmids carrying the MCR-1 gene (*Liu et al., 2016*; *Sun et al., 2018*; *Wang et al., 2018*). MCR-1 is selectively advantageous in the presence of colistin (*Liu et al., 2016*; *Sun et al., 2018*), but it generates a colistin resistance phenotype that is of borderline clinical significance (*Liu et al., 2016*; *Bardet et al., 2017*; *Doumith et al., 2016*; *Yang et al., 2017*). In our experiments, MCR-1 populations consistently evolved high-level colistin resistance by acquiring mutations in *lpxC*, an essential gene involved in lipopolysaccharide (LPS) biosynthesis, that interacts epistatically with MCR-1 to increase resistance. Wild-type populations, on the other hand, were unable to evolve increased colistin resistance under the strong selective regime used in our experiments. We then tested for associations between MCR-1 and *lpxC* polymorphisms in pathogenic and commensal *E. coli* using large genomic datasets to better understand the consequences of this interaction in vivo.

## Results

### Accelerated evolution of colistin resistance in MCR-1 *E. coli*

To search for genes that interact with MCR-1 to increase colistin resistance, we evolved populations of *E. coli* carrying MCR-1 encoded on an IncX4 plasmid (MCREC) and wild-type control populations under strong selection for increased colistin resistance. Our selection experiment used an 'evolutionary ramp' in which the concentration of colistin was doubled daily from a very low dose (1/8 minimum inhibitory concentration [MIC]) to a very high dose (16× MIC) (see Methods). Antibiotic doses were standardized relative to the MICs of the two parental strains to ensure that the strength of selection was equivalent for the two strains. The rapid increase in antibiotic concentration in the 'evolutionary ramp' treatment ensures that populations must either evolve increased resistance or face extinction (once concentrations exceed the MIC of the parental strains) (*Gifford et al., 2018*; *San Millan et al., 2016*). Thus, the rate of populations extinction defines the measure of the evolvability of a strain.

Previous selection experiments have shown that mutants with low resistance tend to have high resistance evolvability, generating characteristic patterns of diminishing returns adaptation (*MacLean*

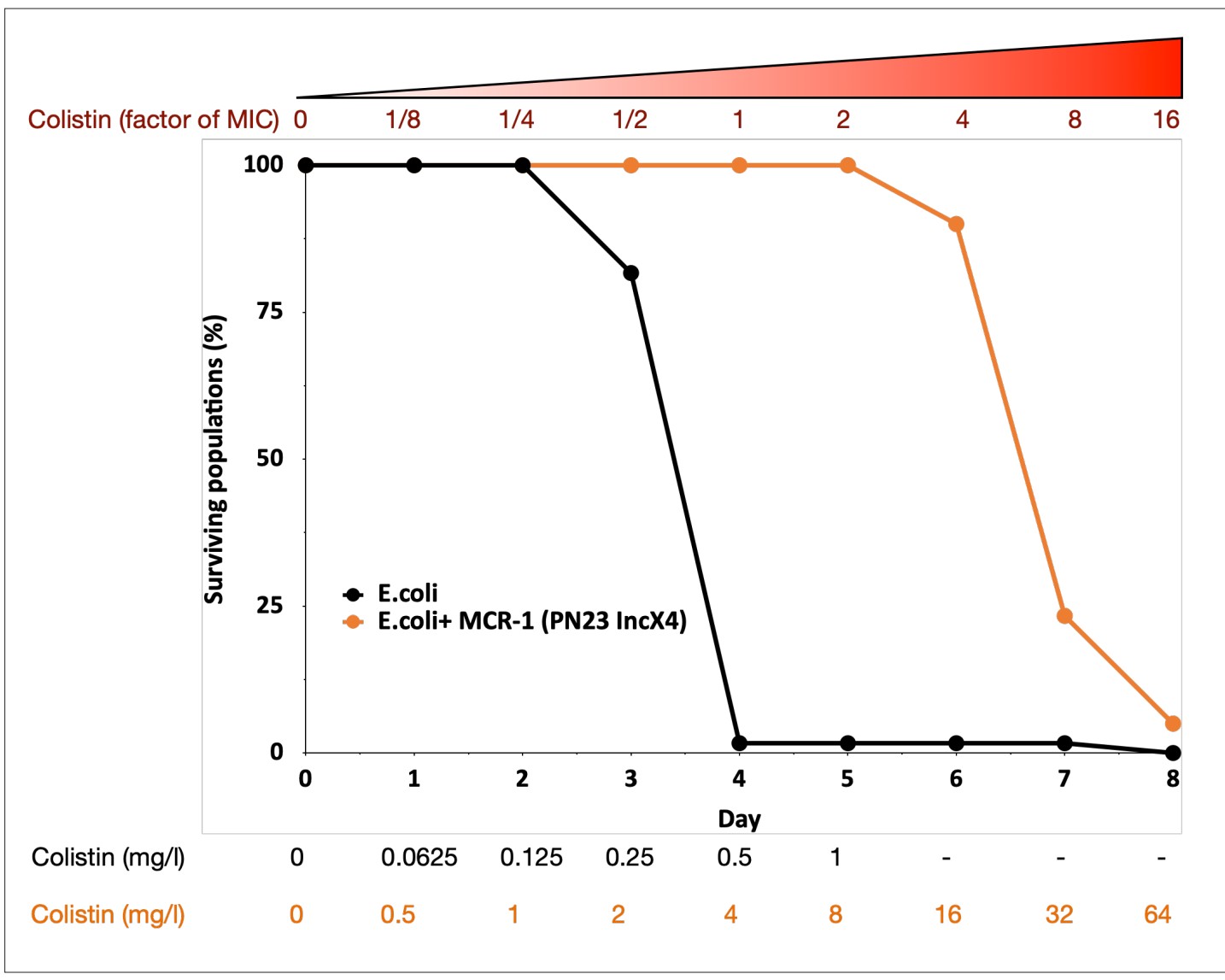

**Figure 1.** MCR-1 carrying bacterial populations (MCREC) show accelerated evolution of high-level colistin resistance. The plot shows the survival of wild-type and MCR-1 carrying populations of *E. coli* (*n* = 120) populations for *E. coli* and *n* = 60 populations for *E. coli* + MCR-1 (PN23 IncX4) over time under increasing selection for colistin resistance. MCR-1 carrying populations showed increased survival, which is indicative of higher colistin evolvability. The numbers shown at the bottom represent the colistin concentration for wild-type *E. coli* (black) and for *E. coli* with MCR-1 (orange).

*et al., 2010*) that has been reported for other microbial phenotypes (*Kryazhimskiy et al., 2014*; *Wiser et al., 2013*). All else being equal, we would therefore expect that control populations (initial MIC = 0.5mg/l) would have higher evolvability than the MCREC populations (initial MIC = 4mg/l). In contrast to this expectation, MCREC populations showed consistent evidence of adaptation to colistin, whereas only a very small minority of control populations were able to evolve increased resistance (*Figure 1*; proportional hazard likelihood ratio = 116, p = <0.0001).

## Mechanistic basis of the evolution of high-level colistin resistance

The increased evolvability of MCREC populations suggests that the presence of MCR-1 opened up new pathways to evolving elevated colistin resistance that were not accessible to wild-type populations. To test this idea, we sequenced the genomes of *n* = 21 independently evolved MCREC clones from day 6 of the experiment (4× MIC). All of the evolved clones had increased colistin resistance (mean MIC = 19.2mg/l, range = 8–32mg/l) compared to the MCREC parental strain (MIC = 4mg/l) (*Supplementary file 1*).

Strikingly, the majority of MCREC clones (*n* = 11/21) carried mutations in LpxC (*Figure 2a*), an UDP-3-*O*-acyl-*N*-acetylglucosamine deacetylase that catalyses the first step in lipid A biosynthesis and is important for membrane composition and asymmetry (*Guest et al., 2021*; *Fivenson and Bernhardt, 2020*). *lpxC* is an essential gene (*Beall and Lutkenhaus, 1987*), suggesting that its evolution should be constrained. Consistent with this idea, *lpxC* mutations were clustered in the coding sequence and both of the observed deletions kept the coding sequence in frame. Interestingly, several clones had mutations in the metal- (zinc) binding domain site which is very important for *lpxC* activity (*Jackman et al., 2001*; *Figure 2b*). For example, mutations at positions 79, 238, 242, and 246 lead to a hundred to thousand-fold decrease in *lpxC* activity (*Jackman et al., 2001*; *Clayton et al., 2013*; *Lee et al., 2014*), and many of our mutations map closely to these important sites (*Figure 2b*). Mutations in other genes involved in LPS biosynthesis were found in evolved clones that lacked *lpxC* mutations, including *lpxA* (*n* = 2) and *rfaY* (*n* = 1) (*Figure 2a*). Interestingly, a minority of clones lacked mutations and/or structural variants (*n* = 4). The colistin resistance of these clones did not differ suggesting that adaptive plasticity in gene expression and/or unstable genetic changes (i.e., heteroresistance; *Andersson et al., 2019*) can also contribute to increased colistin resistance (*Figure 2—figure supplement 1*).

## Testing for interactions between LpxC and MCR-1

The presence of *lpxC* mutations in MCREC populations suggests that the presence of MCR-1 increases the selective advantage provided by *lpxC* mutations in the presence of colistin. To test this hypothesis, we focussed on measuring the interaction between MCR-1 and a *lpxC* mutation from a domain that is critical for LpxC function (i.e., I244N). The MCR-1 natural plasmid used in our experiments (i.e., IncX-4:MCR-1) is an intermediate size plasmid (approx. 34kb) that also carries genes for housekeeping and conjugation, as well as cargo genes of unknown function (*Yang et al., 2017*). To examine the interaction between LpxC and MCR-1 more directly, we cloned MCR-1 into pSEVA121, a low copy number (approx. 5 copies/cell) synthetic plasmid that confers a similar level of colistin resistance (MIC = 5mg/l) to the IncX4:MCR-1 plasmid used the selection experiment (MIC = 4mg/l). The key advantage of the pSEVA system is that this makes it possible to measure the impact of realistic levels of MCR-1 expression while controlling for any background plasmid effects using an empty vector control.

Our approach to measure the interaction between LpxC and MCR-1 was to test for epistasis between these genes in quantitative traits that were key targets of selection in our experiment. Epistasis occurs when different genes make a non-additive contribution to phenotypes (*Durão et al., 2018*). For any quantitative trait, epistasis (*e*) can be calculated as $e = w_{AB}*w_{ab} - w_{Ab}w_{aB}$ where $w_{AB}$ is the value of the trait in the wild type, ab is the double mutants, and the single mutants are represented as Ab and aB (*Trindade et al., 2009*). Negative epistasis (*e* < 0) occurs when the value of the trait is less in the double mutant than would be expected from the single mutant, and vice versa for positive epistasis (*e* > 0). Although evolutionary biologists typically focus on epistatic interactions for fitness, the same methods can be applied to calculate epistasis for any quantitative trait.

The rapid increase in colistin concentration in our experiment implies that colistin resistance is likely to have been a key target of selection. MCR-1 increased the colistin resistance of the *E. coli* parental strain from 0.5 to 4mg/l, which is the clinical breakpoint MIC for colistin resistance (*The European Committee on Antimicrobial Susceptibility Testing, 2022*). The *lpxC* mutation, on the other hand,

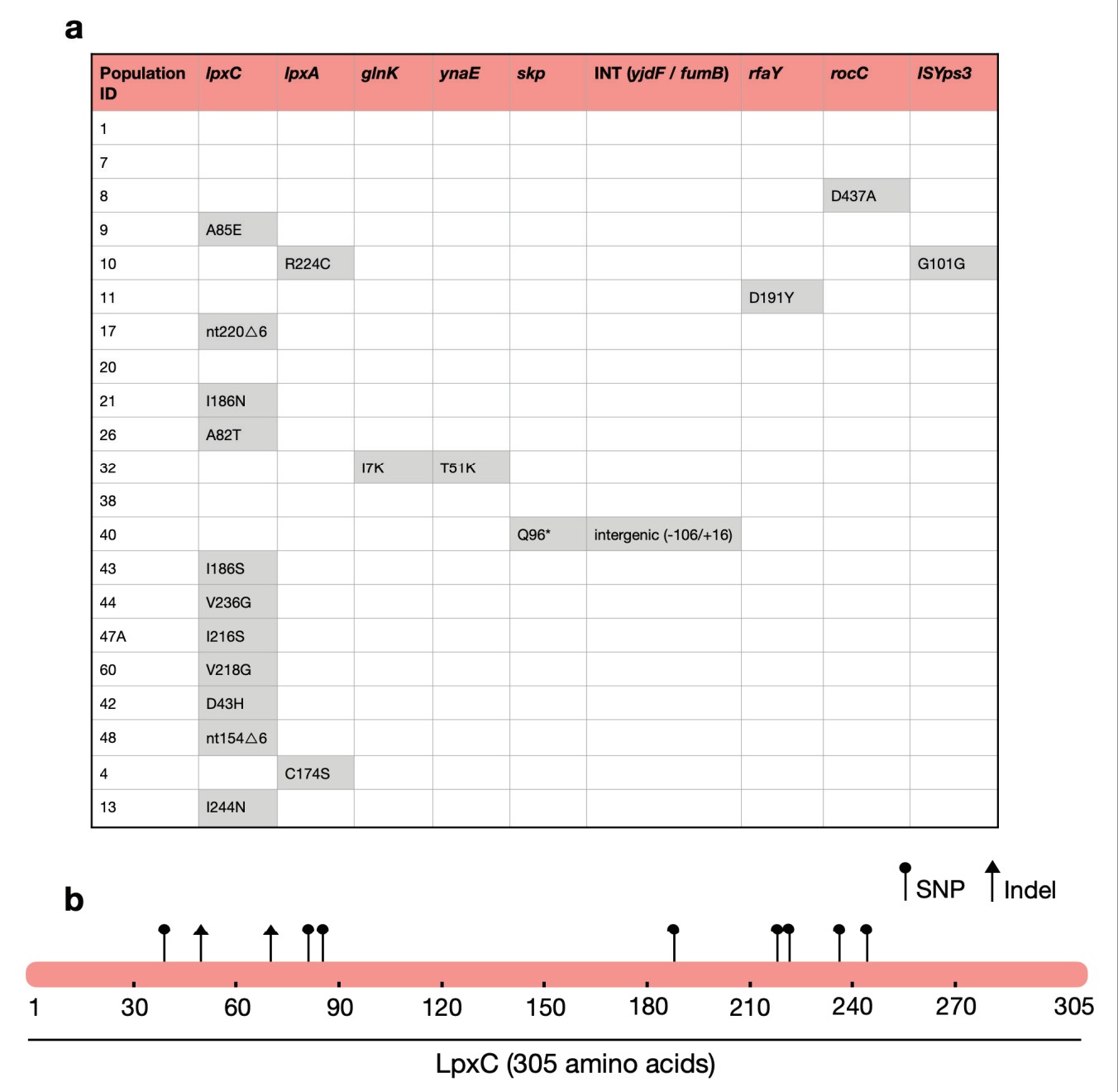

**Figure 2.** The majority of MCREC clones show mutations in LpxC.

(**a**) List of mutations identified in colistin-evolved MCR-1-positive clones. (**b**) Figure shows the mutations identified in LpxC. LpxC positions 79, 238, 242, and 246 are important for Zn binding and LpxC activity (*Jackman et al., 2001*). nt – nucleotide; △ – deletion; SNP – single-nucleotide polymorphism; Indel – insertions/deletions of nucleotides.

The online version of this article includes the following figure supplement(s) for figure 2:

**Figure supplement 1.** Figure comparing the colistin resistance of clones with mutations in *lpxA/C*, other targets, and no mutations (i.e. none).

**Figure supplement 2.** Sequencing reads of each population were mapped to the hybrid assembly of reference d0-5-1.

**Figure supplement 3.** Sequencing coverage (*y*-axis) of each coding region (black dots) in the chromosome of each population.

**Figure supplement 4.** Sequencing coverage (*y*-axis) of each coding region (black dot) in the PN23 plasmid of each population.

marginally decreased colistin resistance in the wild-type strain (~35% decrease in MIC) (*Figure 3a*). However, the combination of *lpxC* and MCR-1 increased colistin resistance to about 2× higher than the levels observed in the MCR-1 strain, revealing positive sign epistasis between *lpxC* and MCR-1 for colistin resistance (*Figure 3a*).

Electrostatic attractions occur between colistin, which is positively charged, and the bacterial outer membrane, which is negatively charged. MCR-1 increases resistance to colistin in part by decreasing the net negative charge on the bacterial outer membrane by modifying LPS (*Sabnis et al., 2021*). Given that *lpxC* is involved in LPS biosynthesis, we reasoned that *lpxC* and MCR-1 may interact epistatically to reduce membrane surface charge. As expected, the MCR-1 gene led to a substantial reduction in cell surface charge (*Figure 3b*). The *lpxC* mutation led to a small reduction in net cell surface charge that was not statistically significant (Wilcoxon rank-sum test p = 0.150). However, the combination of the *lpxC* mutation and MCR-1 reduced the surface charge to a greater extent than the MCR-1 alone (*Figure 3b*). The surface charge of the *lpxC*-MCR-1 double mutant was marginally less than the additive expectation, providing evidence of weak negative epistasis for cell surface charge ($e = -0.059$, error = 0.0355).

Antibiotic resistance is associated with costs that modulate the selective benefit of resistance (*Andersson and Hughes, 2010*). Previous work has shown that MCR-1 expression is associated with fitness costs (*Yang et al., 2017*), and it is possible that *lpxC* mutations provide a fitness advantage by compensating for the costs of MCR expression. To address this possibility, we measured the fitness of mutants in antibiotic-free culture medium. As expected, MCR-1 was associated with a fitness cost (*Figure 3c*). *lpxC* is an essential gene and extensive pleiotropic effects of *lpxC* mutations have been reported, including defective cell division, altered membrane composition, and leakage of periplasmic enzymes (*Young and Silver, 1991*; *Kloser et al., 1996*), suggesting that mutations in *lpxC* should be deleterious. Consistent with this idea, the *lpxC* mutation was associated with a high fitness cost in both the presence and absence of MCR-1, allowing us to rule out the possibility that *lpxC* mutations represent a compensatory adaptation for MCR-1 (*Figure 3c*). Interestingly, the fitness of double mutant was marginally less than we would expect under an additive model, revealing weak negative epistasis for fitness ($\varepsilon = -0.0244$, error = 0.0089).

## Association between MCR-1 and LpxC in pathogenic *E. coli*

The large-scale use of colistin as a growth promoter in agriculture drove the spread of MCR-1 across agricultural, environmental, and human-associated populations of *E. coli* in South East Asia (*Sun et al., 2018*; *Wang et al., 2018*). To test the importance of *lpxC* in colistin resistance, we searched for *lpxC* polymorphisms in a recently published dataset of MCREC isolates from China collected from a diversity of agricultural, environmental, and human-associated sources (*Shen et al., 2020*). Notably, the frequency of *lpxC* polymorphisms in pathogenic *E. coli* isolates collected from infections was higher than the level found in patients colonized with *E. coli* and in the environment (*Figure 4—figure supplement 1*). Colistin is used as a last line of defence antibiotic for the treatment of infections caused by multidrug-resistant pathogens (*Li et al., 2006*) and this association between *lpxC* and MCR-1 in pathogenic *E. coli* is worrying, as it suggests that high-level colistin resistance may be common in pathogenic MCR-1-positive isolates.

Although this association between MCR and *lpxC* mutations in pathogenic *E. coli* appears compelling, the population structure of *E. coli* differs between isolate sources. This is an important issue to address, as it is not clear a priori if the high diversity of *lpxC* in pathogenic *E. coli* reflects the high diversity of this niche as opposed to the high diversity of particular phylogroups. In particular, extraintestinal pathogenic *E. coli* usually belong to phylogroup B2 and D (*Picard et al., 1999*; *Johnson et al., 2001*; *Moulin-Schouleur et al., 2007*), with B2 the main lineage for the majority of bloodstream *E. coli* infections in the UK (*Kallonen et al., 2017*). Similarly, the majority of B2 isolates in the Shen dataset were from patients infected with MRCEC (*n* = 10/18). It is not clear whether these *lpxC* mutations in pathogenic MCREC were acquired after MCR-1 (as in our experiments) or if these are pre-existing *lpxC* polymorphisms that were present in the B2 phylogroup prior to the acquisition of MCR-1. To address this issue, we analysed *lpxC* diversity in genomic datasets of commensal, pathogenic, and animal-associated *E. coli* from the UK, where MCR-1 is effectively absent (*Davies et al., 2020*; *Shaw et al., 2021*). The diversity of *lpxC* was consistently highest in isolates from the B2 phylogroup in this MCR-1 naive population of *E. coli* (Figure 5), suggesting that high levels of *lpxC* in B2 pre-date the

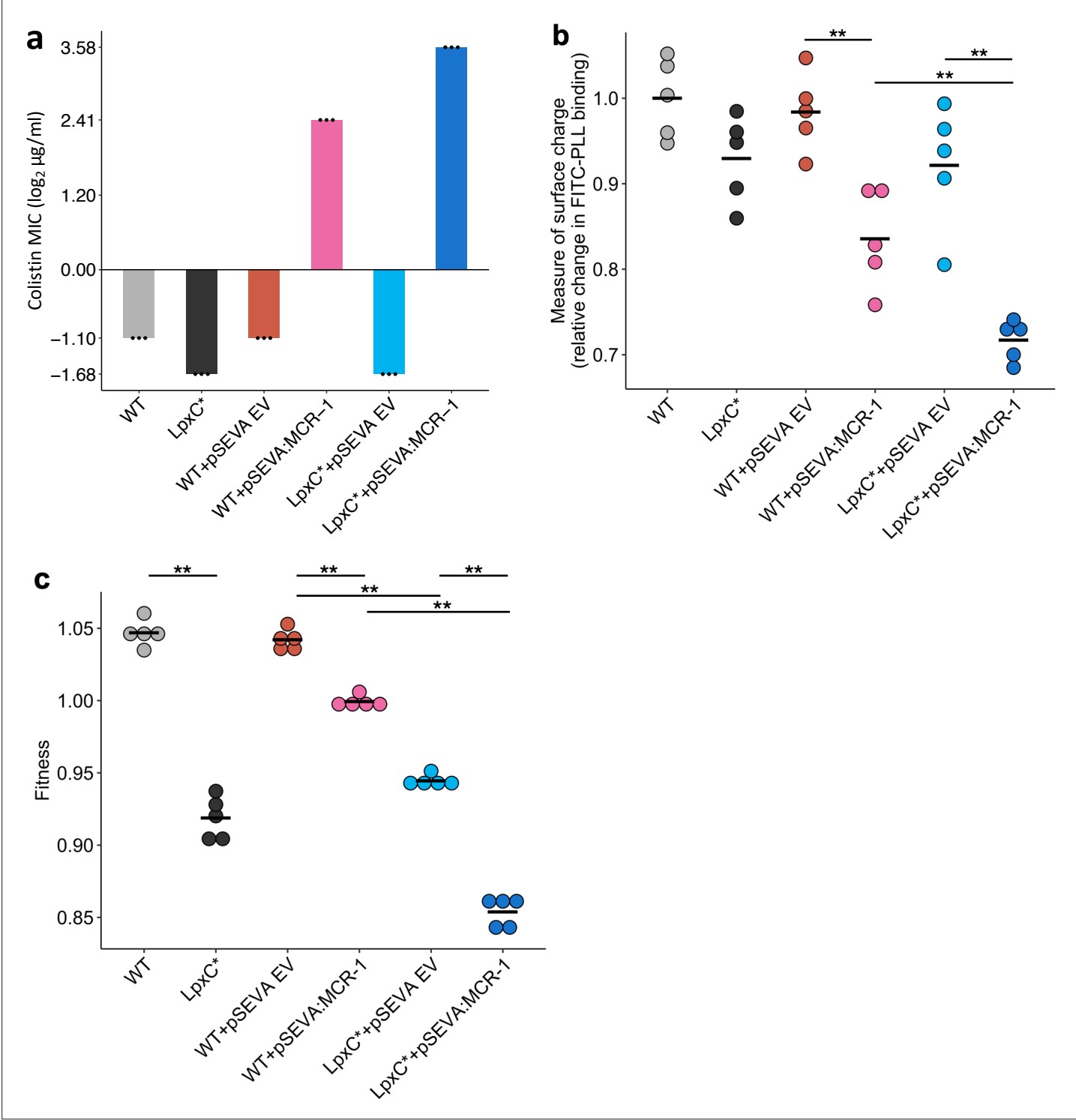

**Figure 3.** Effect of the *lpxC* mutation (*lpxC\**), with and without MCR-1, on bacterial susceptibility to colistin resistance (**a**), membrane surface charge (**b**), and fitness (**c**). (**a**) Bacterial susceptibility to colistin was tested by measuring minimum inhibitory concentration (MIC) (*n* = 3 biological replicates per strain). (**b**) Relative change in membrane surface charge was measured by using fluorescein isothiocyanate-labelled poly-L-lysine (FITC-PLL)-binding assay (*n* = 5 biological replicates/strain) (see Methods). The changes were compared to wild-type (WT) control strain. (**c**) Bacterial fitness was measured using in vitro competition assay where strains were competed against a tester *E. coli* strain carrying a chromosomally integrated GFP (*n* = 5 biological replicates per competition). Statistical significance was determined by pairwise comparisons using the Wilcoxon rank-sum test, and double asterisks show differences with a p value <0.01. Central horizontal lines represent mean values of biological replicates. WT – wild type; EV – empty vector.

acquisition of MCR-1 plasmids by HGT, and that this phylogroup is effectively primed to evolve high-level colistin resistance by MCR-1 plasmid acquisition.

The vast majority of SNPs seen in *lpxC* were synonymous ($n = 6746/6783$), suggesting that *lpxC* is under strong purifying selection. Only 37/1429 isolates had non-synonymous alleles. Strikingly, the majority were within B2 from the Davies infection dataset (33/37, 89%), despite B2 making up only around half of the filtered dataset ($n = 659$, 46%). This greater proportion of isolates with non-synonymous SNPs in B2 compared to other phylogroups was significant (chi-squared test p < 0.001, *X*-squared = 26.7; 33/659 B2 vs. 44/771 non-B2). We verified this higher diversity of *lpxC* in phylogroup B2 using complete RefSeq assemblies (*Figure 5—figure supplement 1*). However, RefSeq isolates did not show a greater proportion of non-synonymous SNPs in B2, with both having ~2% of isolates with non-synonymous SNPs (chi-squared test p = 1, *X*-squared <0.001; 6/278 B2 vs. 35/1611 non-B2). It may be relevant that the Davies dataset was from bloodstream infections whereas RefSeq includes a diverse range of isolates.

The greater diversity of *lpxC* in the pathogenic B2 phylogroup across studies is intriguing. Levels of overall genetic diversity are known to differ between *E. coli* phylogroups, with phylogroup D having the highest nucleotide diversity across 112 core *E. coli* gene families in a recent study (*Touchon et al., 2020*). However, phylogroup B2 was found to have similar overall nucleotide diversity to major phylogroups such as A or B1, suggesting that our observation here of higher diversity in *lpxC* from phylogroup B2 is not simply an artefact of higher overall genetic diversity. Altogether, this analysis suggests that pathogenic populations of *E. coli* already contain higher levels of standing genetic variation in *lpxC*, which may pre-dispose them to evolving high-level colistin resistance by acquiring conjugative plasmids carrying MCR-1.

As a final test for the importance of interactions between *lpxC* and MCR-1 in clinical settings, we compared the mutations observed in our evolved clones with those found in B2 isolates. We found no shared mutations between the clinical isolates and the lab-evolved isolates, either at the level of individual nucleotides or codons (*Figure 5—figure supplement 2*). However, it is unclear if we should expect to see an overlap for two reasons.

First, the mutations found in clinical isolates likely reflect a combination of beneficial mutations, neutral mutations, and weakly deleterious mutations across multiple strains, whereas the mutations found in our isolates reflect mutations that were beneficial under colistin treatment in a single MCR-1 carrying strain. Given this, it is unclear to what extent we should expect to see a strong overlap of mutated sites, but this is nonetheless an interesting comparison to make.

Second, our data suggest that a large number of mutations in *lpxC* can interact with MCR-1 to increase colistin resistance. Experimental evolution studies have shown that while parallel evolution at the level of the gene is quite common, parallelism at the level of individual nucleotides is rare (*Stern, 2013*; *Tenaillon et al., 2012*; *Schenk et al., 2022*). The notable exception to this is cases where strong selective constraints limit the potential number of adaptive mutations, as has been observed for hotspot mutations in DNA gyrase and topoisomerase that are associated with fluoroquinolone resistance (*Papkou et al., 2020*; *Wong et al., 2012*). In our evolution experiment, we found no cases of parallel evolution at the nucleotide level and only a single example of codon-level parallelism (I86 S/N), giving an estimate of the probability of codon-level parallelism of ($1/9^2 = 1/81$). This low probability of parallel evolution implies that a very large number of mutations in *lpxC* are likely to interact epistatically with MCR-1 to increase resistance. Given this, it is perhaps not surprising that we did not observe any overlap between lab-evolved mutations and naturally occurring polymorphisms in *lpxC*. Furthermore, *lpxC* mutations will have many other impacts beyond colistin resistance. It seems plausible that pathogenic *E. coli* may have higher *lpxC* diversity due to the different host environment they may have recently inhabited compared to non-pathogenic *E. coli*.

## Discussion

The acquisition of specialized antibiotic resistance genes through horizontal gene transfer has played a key role in the evolution of antibiotic resistance in pathogenic bacteria. One of the key insights from genomics studies has been that human pathogens are highly polymorphic (*Baker et al., 2018*; *Didelot et al., 2012*), but the role of chromosomal genetic variation in shaping the acquisition and fitness effects of mobile resistance genes remains poorly understood (*Bottery et al., 2017*; *McNally et al., 2016*; *San Millan et al., 2015*). Here, we show that the evolution of high-level colistin resistance

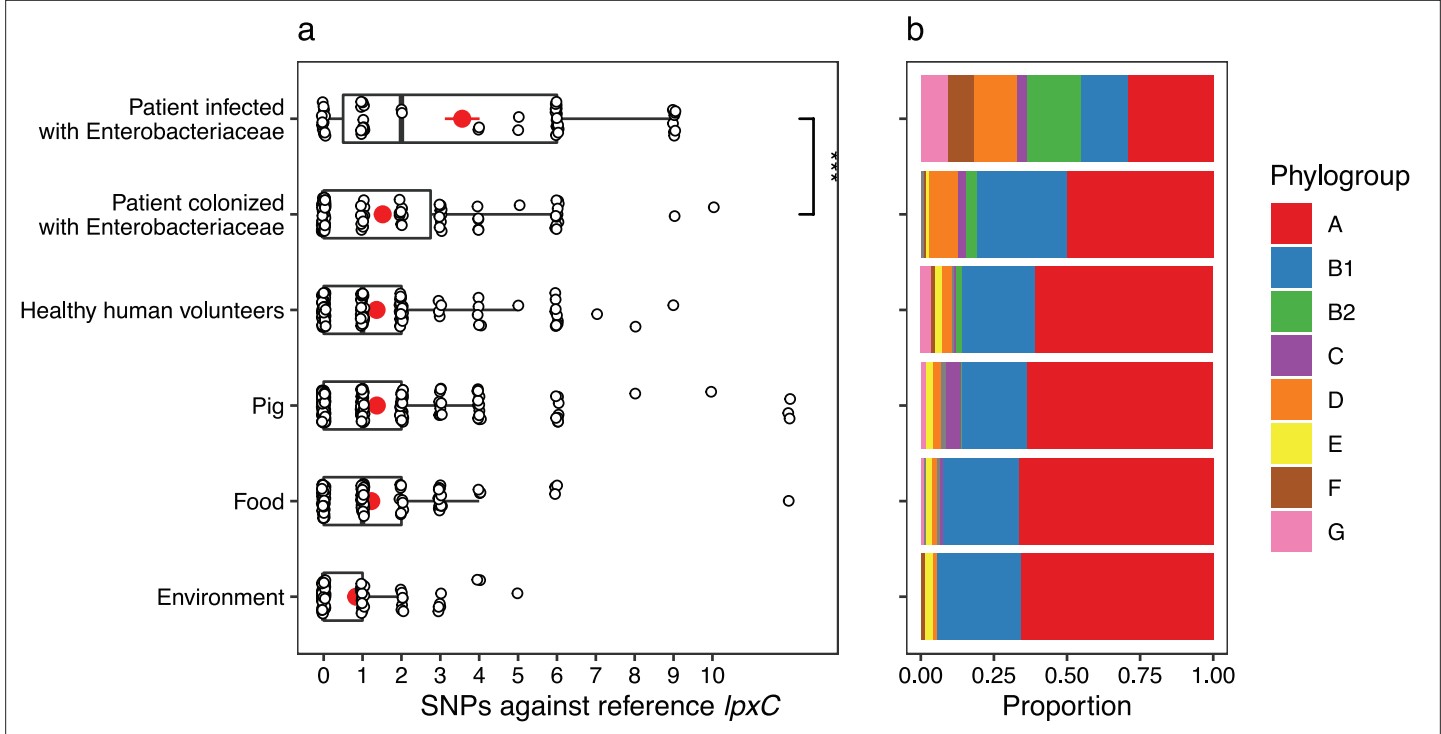

**Figure 4.** *mcr-1*-positive isolates from patients infected with Enterobacteriaceae have more mutations in *lpxC* compared to isolates from other sources. Data shown are from *n* = 673 unfiltered MCR-1 plasmid carrying isolates collected in 2016 by ***Shen et al., 2020***. (**a**) Single-nucleotide polymorphisms (SNPs) in *lpxC* determined by the alignment of the *lpxC* sequence for each assembled isolate against the reference *lpxC* sequence from *E. coli* MG1655 (NC_000913.3:106557–107474). Mean values (± standard error) are shown by red points (lines). Isolates from patients infected with Enterobacteriaceae have significantly more SNPs than other sources, for example, those from patients only colonized with Enterobacteriaceae (Wilcoxon rank-sum test, *W* = 1150, p < 0.001). ***p < 0.001. However, this difference is associated with (**b**) the population structure of *E. coli*, with a greater proportion of phylogroups B2 and D in isolates from infected patients. The same plot for filtered isolates (***Figure 4—figure supplement 1***) showed a similar pattern although the comparison between patients infected vs. colonized was not significant.

The online version of this article includes the following figure supplement(s) for figure 4:

**Figure supplement 1.** A version of ***Figure 4*** showing single-nucleotide polymorphisms (SNPs) in *lpxC* in the Shen dataset with filtering criteria for isolate assemblies applied (see Methods).

is driven by epistatic interactions between the MCR-1 mobile colistin resistance and *lpxC*, a chromosomal gene involved in LPS biosynthesis (***Figures 1–3***). *lpxC* mutations are more common in pathogenic MCR-positive *E. coli* (MCREC) than commensal MCREC, highlighting the clinical relevance of this interaction (***Figure 4***). Crucially, *lpxC* polymorphisms are over-represented in pathogenic lineages for *E. coli* from populations with no history of MCR-1 acquisition, suggesting that pre-existing polymorphisms in *lpxC* potentiated the evolution of high-level colistin resistance through the acquisition of conjugative plasmids carrying MCR-1 (***Figure 5***). The underlying causes of high *lpxC* diversity in pathogenic *E. coli* are unclear, and we hope that future work will shed light on the impact of *lpxC* polymorphisms on *E. coli* virulence and infection.

One of the most surprising features of our results is that approximately 20% of clones that were recovered from MCR-1-positive populations growing at 4× MIC (16mg/l) did not have any detectable mutations or structural variants, a phenomenon that we have also reported in *Pseudomonas aeruginosa* (***Kapel et al., 2022***). There are a number of examples of adaptation to colistin through the acquisition of unstable resistance mutations that rapidly revert when selection for resistance is relaxed (i.e. heteroresistance ***Andersson et al., 2019***). Importantly, we cultured clones for DNA extraction on a medium containing a low dose of colistin (1mg/l, ¼ MIC) that may not have generated effective selection for the maintenance of unstable resistance determinants. Alternatively, it is possible that these clones were sampled from populations that adapted to colistin phenotypically, for example, due to the growth of a subset of cells with altered expression of genes involved in LPS biosynthesis. Given the importance of colistin as a 'last line of defence' antibiotic, an important goal for future work will

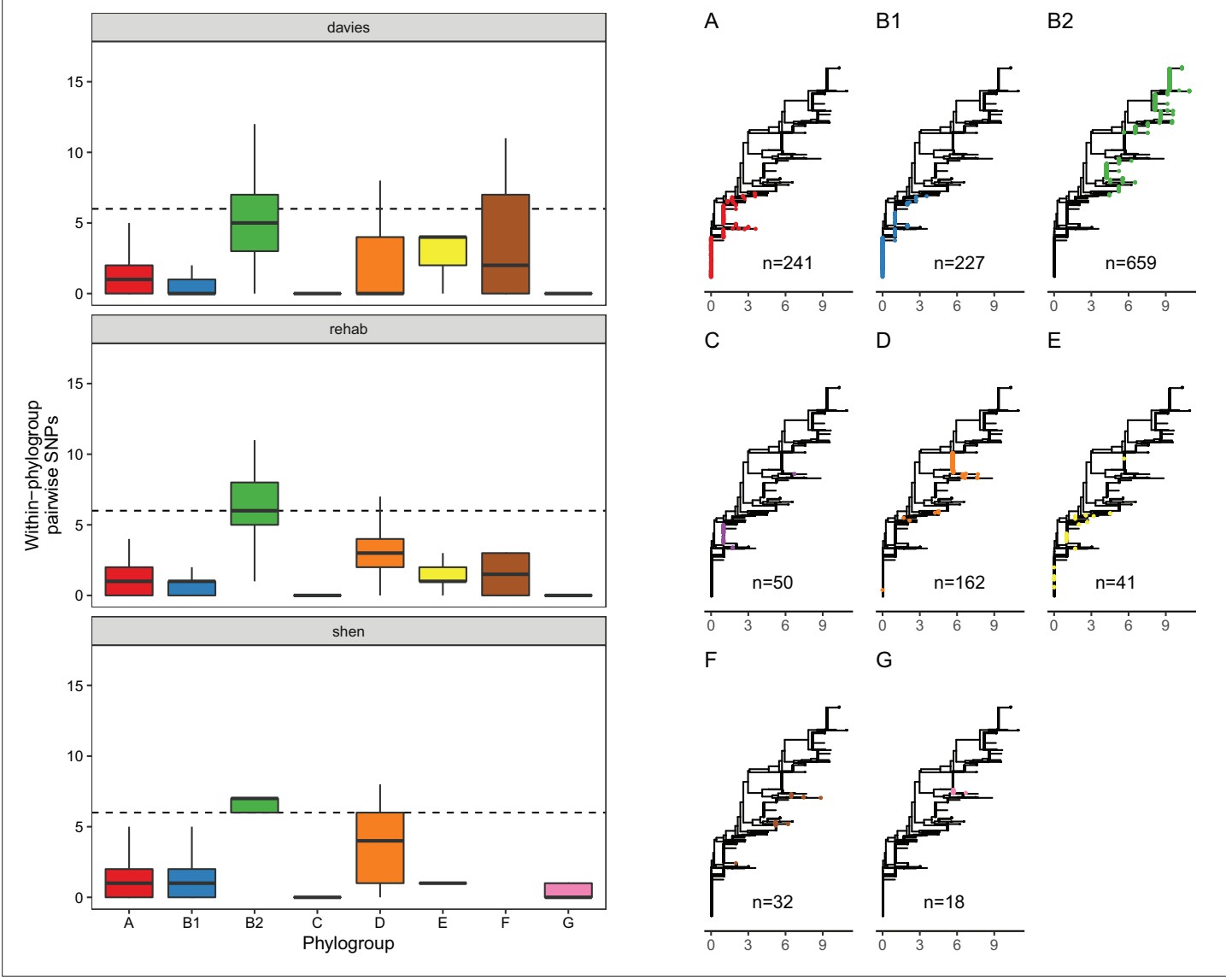

**Figure 5.** LpxC diversity in main *E. coli* phylogroups. The diversity of *lpxC* within phylogroup B2 is greater across three different datasets (called ***Davies et al., 2020***, ***Shaw et al., 2021***, ***Shen et al., 2020***) (see Methods for details). Left: boxplots of the pairwise single-nucleotide polymorphisms (SNPs) between *lpxC* sequences of the same phylogroup within different studies. Dashed line shows the median pairwise distance (six SNPs) between isolates of different phylogroups. Right: a midpoint-rooted neighbour-joining tree of *lpxC* diversity, with coloured tips in each panel showing the location of isolates from each main phylogroup (A–G). Assembled isolates were filtered to only include high-quality assemblies (see Methods).

The online version of this article includes the following figure supplement(s) for figure 5:

**Figure supplement 1.** Analysing NCBI RefSeq complete assemblies shows that B2 isolates have more single-nucleotide polymorphisms (SNPs) relative to the reference *lpxC* than other phylogroups, reproducing the finding from the three assembled datasets (***Figure 5***).

**Figure supplement 2.** Correlation between the sites mutated in public genomes and the experimentally mutated sites (blue), as well as the four sites involved in metal binding of *lpxC* (red).

be to investigate the mechanisms that underpin revertible colistin resistance and how these interact with MCR-1.

There is growing evidence that standing genetic variation (i.e., pre-existing variation) in pathogen species has played an important role in the evolution of antibiotic resistance. For example, at least two of the major lineages of methicillin-resistant *S. aureus* originated well before the clinical introduction of methicillin (***Harkins et al., 2017***; ***Gill et al., 2021***). Our study builds on previous work (***Vogwill et al., 2014***; ***Bethke et al., 2020***; ***Papkou et al., 2020***; ***Gifford et al., 2018***) by demonstrating that

epistatic interactions between mobile resistance genes and chromosomal variants shape the evolution of resistance. One limitation of our study is that we only focussed on interactions between the chromosome and the MCR-1 resistance gene. We suggest that an important goal for future work will be to study the importance of genetic interactions between bacterial chromosomes and mobile elements that transfer AMR genes (*Partridge et al., 2018*) more broadly. We speculate that chromosome–mobile element interactions likely to be particularly important for highly mobile resistance genes, such as MCR-1 (*Liu et al., 2016*; *Wang et al., 2018*; *Wang et al., 2017*) and carbapenemases (*León-Sampedro et al., 2021*) that are rapidly transferred between divergent bacterial strains and species. Our findings suggest that understanding interactions between mobile resistance genes and background genetic variation could help to develop methods for predicting the spread of resistance in pathogen populations, which is an important goal for evolutionary microbiology.

## Methods

### Bacterial strains and growth condition

All the experiments were carried out in *E. coli* J53. Experiments were conducted in Luria-Bertani (LB) medium (Sigma-Aldrich).

### pSEVA:MCR-1 vector construction

A synthetic MCR-1 plasmid was constructed by cloning *mcr-1* gene into pSEVA121 plasmid (*Wong et al., 2012*). The *mcr-1* gene along with its natural promoter was amplified from the natural PN16 (IncI2) plasmid using Q5 High-Fidelity DNA Polymerase (New England BioLabs). The amplified and purified *mcr-1* fragment was assembled together with PCR-amplified pSEVA121 backbone using NEBuilder HiFi DNA Assembly Master Mix according to the manufacturer's instructions. The assembled plasmid was then transformed into *E. coli* J53 using the standard electroporation method. Cells were recovered in 1 ml SOC (super optimal broth with catabolite repression) medium, followed by 1-hr incubation at 37°C. Different dilutions of transformant mixture were made and were plated onto Petri dishes containing LB agar supplemented with 100 µg/ml ampicillin. The culture plates were incubated at 37°C for overnight. PCR and sequence verification by Sanger sequencing were performed to ensure the presence of the correctly assembled recombinant plasmid. A full list of the primers used is given in *Supplementary file 2*.

### Experimental evolution

The experimental evolution experiment was carried out using our previously established 'evolutionary ramp' method (*Gifford et al., 2018*; *San Millan et al., 2016*). Briefly, wild-type *E. coli* (control) and *E. coli* with PN23 IncX4:MCR-1 plasmid were grown on LB agar medium supplemented with appropriate antibiotic at 37°C for overnight. To ensure reproducibility, we had 120 and 60 independent replicates for *E. coli* control and *E. coli*-MCR-1 strain (MCREC), respectively. Parallel cultures were propagated in 96-well microtitre plates where the inner 60 wells were used for bacterial cultures and the outer wells for blank measurement and contamination controls. We used checkerboard layout on the plate to control for potential cross-contaminations. The plates were incubated for 22 hr and then overnight grown cultures were diluted 1:400-fold into fresh LB medium plates with increasing concentration of colistin. The concentration of colistin was doubled daily from the very low dose (1/8 MIC) to a very high dose (16× MIC). At the end of each transfer, the optical density was measured, and the experiment was carried out until more than >90% of cultures have gone extinct (i.e. $OD_{595}$ less than 0.05). In total, eight transfers were performed in LB medium with colistin. The evolved *E. coli*-MCR-1 and control populations were preserved at −80°C in 15% glycerol solution. We tested for a difference between the survival of control populations ($n = 120$) and plasmid carrying populations ($n = 60$) using Cox's proportional hazard model. Each population was considered extinct on the day when OD fell below our 0.05 threshold.

### DNA extraction, whole-genome sequencing, and data analysis

To identify the polymorphisms associated with colistin resistance evolution, we selected 21 evolved populations from day 6 (4× MIC). Bacterial DNA was extracted using DNeasy Blood & Tissue Kit (Qiagen) on a QIAcube automated system (Qiagen). The concentration of each genomic DNA sample

was determined using a QuantiFluor dsDNA System (Promega) following the manufacturer's protocol. The purified genomic DNA samples were subjected to Illumina HiSeq 4000 platform at the Oxford Genomics Centre (Wellcome Centre for Human Genetics, University of Oxford, UK). The ancestral isolate was sequenced on an Oxford nanopore MinION platform using a FLO-MIN106 flow-cell and the SQK-LSK109 Ligation Sequencing kit.

Raw Illumina reads were quality controlled with Trimmomatic v. 0.39 (*Bolger et al., 2014*) using the ILLUMINACLIP (2:30:10) and SLIDING WINDOW (4:15) parameters. Raw nanopore reads were basecalled using guppy v. 0.0.0+7,969d57 and reads were demultiplexed using qcat v. 1.1.0 (https://github.com/nanoporetech/qcat; *Rescheneder, 2019*). Resulting sequencing reads were assembled using unicycler v. 0.4.8 (*Wick et al., 2017*), which used SAMtools v. 1.9 (*Li et al., 2009*), pilon v. 1.23 (*Walker et al., 2014*), and bowtie2 v. 2.3.5.1 (*Langmead and Salzberg, 2012*), in hybrid mode with respective Illumina reads. The resulting assembly was annotated using Prokka v. 1.14.0 (*Seemann, 2014*).

Small variants were identified using breseq v. 0.34.0 (*Deatherage and Barrick, 2014*) in the polymorphism prediction mode, which implemented bowtie2 v. 2.3.5.1. Copy number variants were estimated with the default parameters of the package CNOGpro v. 1.1 (*Brynildsrud et al., 2015*) in R (*R Development Core Team, 2021*). The plasmid copy number was calculated by dividing the average sequencing coverage of each plasmid by the average sequencing coverage of the chromosome using the SAMtools depth algorithm (*Figure 2—figure supplements 2–4*).

## Construction of I244N lpxC mutant (*lpxC\**)

A previously described pORTMAGE method (*Nyerges et al., 2016*) was applied to introduce I244N *lpxC* mutation into the wild-type *E. coli* J53. The mutant was constructed via synthetic ssDNA-mediated recombineering using the pORTMAGE3 plasmid. The MODEST tool was used to design 90 nucleotides long ssDNA oligos that had complementary sequences to the replicating lagging strand with a minimized secondary structure ($\geq -12$ kcal mol$^{-1}$) (*Bonde et al., 2014*). The oligos were ordered from Integrated DNA Technologies (Coralville, IA, USA). For the recombineering, pORTMAGE3 vector was transformed into *E. coli* J53 electrocompetent cells using standard electroporation method. *E. coli* carrying pORTMAGE3 vector was further used for recombineering. pORTMAGE3 was induced at 42°C for 15 min to allow for efficient mutation incorporation and to avoid off-target mutagenesis. 1 µl of the 100 µM mutation carrying oligo was transformed into induced electrocompetent cells. Following the electroporation, cells were recovered in 5 ml Terrific-Broth (TB) media (24 g yeast extract, 12 g tryptone, 9.4 g $K_2HPO_4$, and 2 g $KH_2PO_4$ per 1l of water) and incubated at 30°C for 60 min. After the incubation period, 5 ml of LB medium was added, and cells were further incubated at 30°C overnight. Appropriate dilutions of the cultures were then plated onto LB agar plates to form individual colonies and incubated at 30°C overnight. Individual colonies were screened for the mutation using Sanger capillary-sequencing. Finally, the pORTMAGE3 plasmid was cured from the sequence-verified colony by growing the cells at 42°C for overnight in antibiotic-free LB medium.

A list of oligos used for the mutation construction is given in *Supplementary file 2*.

## Determination of MIC

MICs were determined using a standard serial broth dilution method with a minor modification that we previously optimized to capture small differences in MIC (*Kintses et al., 2019*). Specifically, smaller colistin concentration steps were used (typically 1.5-fold). Ten-step serial dilution of colistin was prepared in fresh LB medium in 96-well microtitre plates. Three wells contained only LB medium to monitor the bacterial growth in the absence of colistin. Bacterial strains were grown in LB medium supplemented with appropriate antibiotic (100 µg/ml ampicillin for *E. coli* pSEVA:MCR-1) at 30°C for overnight. Following overnight incubation, approximately $5 \times 10^5$ cells were inoculated into the each well of the 96-well microtitre plate. Three independent replicates for each strain and the corresponding control were used. The top and bottom row in the 96-well plate were filled with LB medium to obtain the background OD value of the growth medium. Plates were incubated at 30°C with continuous shaking at 250 rpm. After 20–24 hr of incubation, $OD_{600}$ values were measured in a microplate reader (Biotek Synergy 2). After background subtraction, MIC was defined as the lowest concentration of colistin where the $OD_{600}$ <0.05.

## Membrane surface charge measurement

To measure bacterial membrane surface charge, we carried out a fluorescein isothiocyanate-labelled poly-L-lysine- (FITC-PLL) (Sigma) binding assay (*Kintses et al., 2019*; *Rossetti et al., 2004*). FITC-PLL is a polycationic molecule that binds to an anionic lipid membrane in a charge-dependent manner and is used to investigate the interaction between cationic peptides and charged lipid bilayer membranes. The assay was performed as previously described (*Kintses et al., 2019*). Briefly, bacterial cells were grown overnight in LB medium, and then centrifuged, and washed twice with 1× phosphate-buffered saline (PBS) buffer (pH 7.4). The washed bacterial cells were re-suspended in 1× PBS buffer to a final $OD_{600}$ of 0.1. A freshly prepared FITC-PLL solution was added to the bacterial suspension at a final concentration of 6.5 µg/ml. The suspension was incubated at room temperature for 10 min, and pelleted by centrifugation. The remaining amount of FITC-PLL in the supernatant was determined fluorometrically (excitation at 500 nm and emission at 530 nm) with or without bacterial exposure. The quantity of bound molecules was calculated from the difference between these values. A lower binding of FITC-PLL indicates a less net negative surface charge of the outer bacterial membrane.

## In vitro competition assay

We carried out in vitro competition experiment using a flow cytometry-based sensitive and reproducible method developed in our laboratory (*Gifford et al., 2018*; *San Millan et al., 2016*; *Jangir et al., 2022*). Flow cytometry was performed on an Accuri C6 (Becton Dickenson, Biosciences, UK). The test strains were competed against a GFP-labelled *E. coli* strain J53 to measure the relative fitness. All competitions were carried out in LB medium with five biological replicates per strain, as previously described (*San Millan et al., 2016*; *Jangir et al., 2022*). Briefly, the bacterial cells were grown in LB medium for overnight. The overnight grown cultures were washed with filtered PBS buffer. The washed cells were diluted into a fresh LB medium and mixed approximately at 1:1 ratio with GFP-labelled *E. coli* J53. Before starting the competition, the total cell density in the competition mix was around half million cells, as we also used for MIC assay. The initial density of fluorescent and non-fluorescent cells was estimated in the mix using medium flow rate, recoding 10,000 events and discarding events with forward scatter <10,000 and side scatter <8000. After confirming the actual ratio close to 1:1, the competition plates were incubated at 30°C with shaking at 250 rpm. After overnight incubation, the competition mix was diluted in PBS buffer and cell densities were adjusted around 1000 per microlitre. The final density of fluorescent and non-fluorescent cells was estimated in the competition mix. Using the initial and final density of fluorescent and non-fluorescent cells, the relative fitness was calculated as described below:

$$\text{Relative fitness} = \frac{\log_2\left(\frac{d_1\,(\text{non-fluorescent})}{d_0\,(\text{non-fluorescent})}\right)}{\log_2\left(\frac{d_1\,(\text{fluorescent})}{d_0\,(\text{fluorescent})}\right)},$$

where $d_0$ and $d_1$ represent cell density before and after the competition, respectively.

## Analysis of *E. coli* genomes datasets

We downloaded three short-read datasets of *E. coli* genomes to investigate *lpxC* diversity: *mcr-1*-positive *E. coli* from China from a range of sources including human infection isolates, collected 2016–2018 (n = 688; Shen dataset) ( *Shen et al., 2020*); *mcr-1*-negative *E. coli* bloodstream infection isolates from Oxfordshire, UK, collected 2013–2015 (n = 981 short-read datasets from n = 976 isolates; Davies dataset) (*Davies et al., 2020*); and *mcr-1*-negative *E. coli* from UK, mainly from livestock, collected 2017 (n = 502; REHAB dataset) (*Shaw et al., 2021*).

Scripts and data for analysis, including assemblies, are available on figshare (doi:10.6084/ m9.figshare.18591347). In brief, genomes for the Shen and Davies datasets were assembled from short reads downloaded from the NCBI SRA. These reads were filtered with Trimmomatic v0.39 with default parameters (*Bolger et al., 2014*) and assembled with Spades v3.15.3 (*Bankevich et al., 2012*). Genomes from the REHAB dataset were downloaded from previously assembled genomes from hybrid assembly. We screened genomes for a full-length *lpxC* using blastn against *lpxC* from *E. coli* MG1655 (NZ_LR881938.1_cds_WP_000595482.1_97; identical to *lpxC* in strain J53; both phylogroup A). We discarded genomes without a full-length *lpxC* match (n = 5), those where *lpxC* had >12 SNPs compared to the reference suggesting poor assembly (n = 20, all from Shen dataset)

leaving *n* = 2146 isolates to analyse. Details of these isolates including accessions, phylogroup, and SNPs in *lpxC* are given in *Supplementary file 3*. We assigned phylogroup with ClermonTyper and analysed the *n* = 2,127 isolates which were in the main phylogroups (A, B1, B2, C, D, E, F, and G). To visualize *lpxC* diversity we plotted a midpoint-rooted neighbour-joining tree including the reference. After obtaining initial results similar to *Figure 5* (not shown), we subsequently filtered isolates to include only higher-quality assemblies (L90 <100, <999 contigs) to ensure that assembly quality was not driving the patterns, resulting in *n* = 1429 isolates. Numbers given in Results refer to this filtered dataset.

To reproduce our finding of higher *lpxC* diversity in phylogroup B2 on another dataset, we downloaded all complete *E. coli* assemblies on NCBI RefSeq (as of 13 January 2022, *n* = 1903). Of these, *n* = 1887 (>99%) were from the main phylogroups and had a full-length *lpxC* with <13 SNPs against the reference *lpxC*.

## Acknowledgements

This work was supported by grants from the Wellcome Trust (106918/Z/15Z, CM) and the Medical Research Council (MR/S013768/1, TW and CM). LO was supported by the Biotechnology and Biological Sciences Research Council (BBSRC) doctoral training partnership (BB/M011224/1). YQE is supported by Laboratory of Lingnan Modern Agriculture Project NT2021010 and the National Natural Science Foundation of China (31200150). LPS is a Sir Henry Wellcome Postdoctoral Fellow funded by Wellcome (grant 220422/Z/20/Z). We thank Csaba Pal lab (Biological Research Centre, Szeged, Hungary) for providing the pORTMAGE plasmid. We also thank Ákos Nyerges (Harvard University) for the helpful discussion on pORTMAGE.

## Additional information

### Funding

| Funder | Grant reference number | Author |
|---|---|---|
| Wellcome Trust | 106918/Z/15Z | R Craig MacLean |
| Medical Research Council | MR/S013768/1 | R Craig MacLean<br>Tim Walsh |
| Biotechnology and Biological Sciences Research Council | BB/M011224/1 | Lois Ogunlana |
| National Natural Science Foundation of China | 31200150 | Qiue Yang |

The funders had no role in study design, data collection, and interpretation, or the decision to submit the work for publication. For the purpose of Open Access, the authors have applied a CC BY public copyright license to any Author Accepted Manuscript version arising from this submission.

### Author contributions

Pramod K Jangir, Conceptualization, Formal analysis, Investigation, Methodology, Writing – original draft, Writing – review and editing; Qiue Yang, Conceptualization, Methodology, Writing – original draft; Liam P Shaw, Data curation, Formal analysis, Methodology, Writing – review and editing; Julio Diaz Caballero, Data curation, Formal analysis, Methodology; Lois Ogunlana, Rachel Wheatley, Methodology; Timothy Walsh, Data curation, Investigation; R Craig MacLean, Conceptualization, Formal analysis, Supervision, Funding acquisition, Writing – original draft, Project administration, Writing – review and editing

### Author ORCIDs

Pramod K Jangir ⬤ http://orcid.org/0000-0001-8330-0655
Rachel Wheatley ⬤ http://orcid.org/0000-0003-1212-2286
R Craig MacLean ⬤ http://orcid.org/0000-0002-7941-813X

Decision letter and Author response
Decision letter https://doi.org/10.7554/eLife.78834.sa1
Author response https://doi.org/10.7554/eLife.78834.sa2

## Additional files

### Supplementary files
• Supplementary file 1. Colistin minimum inhibitory concentration (MIC) of the evolved populations.
• Supplementary file 2. Primers used in this study.
• Supplementary file 3. Accessions, phylogroup, and single-nucleotide polymorphisms (SNPs) in lpxC gene in *E. coli* isolates.
• MDAR checklist

### Data availability
All data generated or analysed during this study are included in this article and its Supplementary Information. All sequencing data are available in the NCBI Sequence Read Archive, with accession numbers SRR19785165–SRR19785186.

The following dataset was generated:

| Author(s) | Year | Dataset title | Dataset URL | Database and Identifier |
|---|---|---|---|---|
| Jangir PK, Yang Q, Shaw LP, Caballero JD, Ogunlana L, Wheatley R, Walsh T, MacLean RC | 2022 | Pre-existing chromosomal polymorphisms in pathogenic *E. coli* potentiate the evolution of resistance to a last-resort antibiotic | https://www.ncbi.nlm.nih.gov/bioproject/PRJNA851773 | NCBI BioProject, PRJNA851773 |

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
