## [Editor Report]

This paper combines evolution experiments with genomic analysis of environmental samples to study the evolution of colistin resistance in *Escherichia coli*. It highlights the importance of pre-existing genomic variations on the chromosomes of clinical strains in driving the evolution of antibiotic resistance. The results are relevant for clinical and non-clinical microbiologists studying antibiotic resistance to last-resort drugs like colistin. The design of the research is simple and elegant, and the genomic data analysis connects the in vitro findings to the real world.

---

## [Decision Letter]

**Decision letter after peer review:**

Thank you for submitting your article "Pre-existing chromosomal polymorphisms in pathogenic *E. coli* potentiate the evolution of resistance to a last-resort antibiotic" for consideration by *eLife*. Your article has been reviewed by 3 peer reviewers, including Sara Mitri as the Reviewing Editor and Reviewer #1, and the evaluation has been overseen by a Reviewing Editor and Gisela Storz as the Senior Editor. The following individual involved in the review of your submission has agreed to reveal their identity: Camilo Barbosa (Reviewer #2).

Essential revisions:

The choice of experimental conditions and their limitations need to be discussed more thoroughly. In particular, there were suggestions for alternative experimental designs of the ramp experiment: "repeating the evolution experiments with clinical isolates from different phylogroups and varying levels of standing genetic variation"; "If they perform the evolutionary ramp experiment with a strain carrying lpxC mutant strain, will they see faster evolution of high MIC mutants?"; "The authors may replicate the experiment with their low-copy clone of mcr-1 which would make it easier for the authors to have empty vector in WT as a proper control". While we have agreed that redoing the evolutionary experiments with new conditions is beyond the scope of the paper, your design decisions should be added to the discussion.

It would also be important to clarify/discuss the mismatch between the evolution experiment and the clinical dataset. A related point that we would like to see in a revised manuscript is "adding the lpxC variants reported in figure 2 to the trees of figure 4 (right)." This should help to make the data less speculative.

*Reviewer #1 (Recommendations for the authors):*

­– Figure 2: it was unclear to me whether the mutations in lpxC are exclusively found in the MCR-1 positive lines or also in the WT lines. This should be more clearly specified in the figure and the text: How many times were lpxC mutations observed in the control lines?

– Why does the lpxC mutation in WT reduce its fitness even if it reduces the charge of the cell surface?

– Figure 4A, the x-axis label should specify SNPs in lpxC

– Also, for Figure 4, do all these isolates have the MCR-1 plasmid? I later understood that from the methods section, but it would be better to spell it out in the figure caption

– It took me a while (after reading the discussion) to understand the argument behind the diversity of B2 phylogroup. It might be helpful to explicitly hypothesise that this group would be more susceptible to picking up the MCR-1 plasmid under antibiotic selection

*Reviewer #2 (Recommendations for the authors):*

My main concern lies in the potential overinterpretation of the experimental data and how the analysis of the clinical datasets, which albeit can be more complex, do not necessarily align with the experimental data. The experimental data suggest that the presence of MCR-1 facilitates the diversification of lpxC and through epistatic interactions, increases resistance against colistin compared to control groups. The evidence and experimental confirmation for it are compelling and well presented. However, the analysis of the two clinical datasets indicated two major findings: First, standing genetic variation in lpxC, and essential gene, is high regardless of the presence of MCR-1. And second, that this genetic diversity is strongly associated with particular phylogenetic groups. While this doesn't necessarily contradict the experimental data, it at least makes me wonder if the authors could give more information about the ancestral strain. I think having information about its standing genetic variation at lpxC and its place among the phylogroups could make the experimental data more powerful and corroborate the clinical data in a better way. As a suggestion, and realizing this may be beyond the scope of this paper and is rather for a follow-up study, I would consider repeating the evolution experiments with clinical isolates from different phylogroups and varying levels of standing genetic variation, if available or possible.

*Reviewer #3 (Recommendations for the authors):*

The main concern I have with this work is with the correlation that the lpxC variations in clinical data are the same/similar to what authors found in their experiments reported in figure 1. Adding the lpxC variants from the experiments in the trees of figure 4 (right) will make it clear if their conclusion is justified.

---

## [Author Response]

Essential revisions:The choice of experimental conditions and their limitations need to be discussed more thoroughly. In particular, there were suggestions for alternative experimental designs of the ramp experiment: "repeating the evolution experiments with clinical isolates from different phylogroups and varying levels of standing genetic variation"; "If they perform the evolutionary ramp experiment with a strain carrying lpxC mutant strain, will they see faster evolution of high MIC mutants?"; "The authors may replicate the experiment with their low-copy clone of mcr-1 which would make it easier for the authors to have empty vector in WT as a proper control". While we have agreed that redoing the evolutionary experiments with new conditions is beyond the scope of the paper, your design decisions should be added to the discussion.

Thank you for the comment. Following this, we have edited the text in the discussion (lines 349-354, pages 13-14), highlighting this as a limitation.

It would also be important to clarify/discuss the mismatch between the evolution experiment and the clinical dataset. A related point that we would like to see in a revised manuscript is "adding the lpxC variants reported in figure 2 to the trees of figure 4 (right)." This should help to make the data less speculative.

We have addressed this in the revised manuscript. It’s a good suggestion, however, it is unclear if we should expect to see an overlap for two reasons (see lines 278-304, pages 11-12), including the fact the strain we used here (*E. coli* J53) is not from phylogroup B2. To test this, we compared the mutations observed in our evolved clones with those found in B2 isolates. We found that, although our observed mutations map very close to the mutation’s positions found in B2 isolates (Figure 5—figure supplement 2), there are no shared mutations between the clinical isolates and the lab-evolved isolates.

Reviewer #1 (Recommendations for the authors):­– Figure 2: it was unclear to me whether the mutations in lpxC are exclusively found in the MCR-1 positive lines or also in the WT lines. This should be more clearly specified in the figure and the text: How many times were lpxC mutations observed in the control lines?

Given the low rate of evolution in the wild-type lines, we only sequenced evolved clones from the MCR carrying lines (the low number of WT replicates that evolved resistance would have made it very difficult to draw any meaningful conclusions about similarities and/or differences in the genomic response of WT and MCR-positive populations). This is clearly stated in the revised manuscript (line 134, page 6). We have changed the figure legend to emphasize that all of the mutations were from MCREC (line 128, page 5).

– Why does the lpxC mutation in WT reduce its fitness even if it reduces the charge of the cell surface?

*lpxC* is an essential gene and mutations in *lpxC* cause several pleiotropic effects, such as defects in cell division, altered membrane composition, and leakage of periplasmic enzymes (PMID: 4903169, 1904854, and 8752330). These effects, in turn, can lead to reduced fitness. In the revised text we emphasize that lpxC is an essential gene; hence our expectation is that lpxC mutations will reduce fitness (lines 196-198, pages 7-8).

– Figure 4A, the x-axis label should specify SNPs in lpxC

Done.

– Also, for Figure 4, do all these isolates have the MCR-1 plasmid? I later understood that from the methods section, but it would be better to spell it out in the figure caption

Done (line 230, page 9).

– It took me a while (after reading the discussion) to understand the argument behind the diversity of B2 phylogroup. It might be helpful to explicitly hypothesise that this group would be more susceptible to picking up the MCR-1 plasmid under antibiotic selection

We have revised this section of the text to make this expectation more explicit (lines 254-256, page 10).

Reviewer #2 (Recommendations for the authors):My main concern lies in the potential overinterpretation of the experimental data and how the analysis of the clinical datasets, which albeit can be more complex, do not necessarily align with the experimental data. The experimental data suggest that the presence of MCR-1 facilitates the diversification of lpxC and through epistatic interactions, increases resistance against colistin compared to control groups. The evidence and experimental confirmation for it are compelling and well presented. However, the analysis of the two clinical datasets indicated two major findings: First, standing genetic variation in lpxC, and essential gene, is high regardless of the presence of MCR-1. And second, that this genetic diversity is strongly associated with particular phylogenetic groups. While this doesn't necessarily contradict the experimental data, it at least makes me wonder if the authors could give more information about the ancestral strain.

Thank you for your comment. Our ancestral strain is *E. coli* J53*,* a derivative of *E. coli* K12, which belongs to phylogroup A (PMID: 1699928).

I think having information about its standing genetic variation at lpxC and its place among the phylogroups could make the experimental data more powerful and corroborate the clinical data in a better way. As a suggestion, and realizing this may be beyond the scope of this paper and is rather for a follow-up study, I would consider repeating the evolution experiments with clinical isolates from different phylogroups and varying levels of standing genetic variation, if available or possible.

A very good idea indeed, however, it’s beyond the scope of the paper.

Reviewer #3 (Recommendations for the authors):The main concern I have with this work is with the correlation that the lpxC variations in clinical data are the same/similar to what authors found in their experiments reported in figure 1. Adding the lpxC variants from the experiments in the trees of figure 4 (right) will make it clear if their conclusion is justified.

This is an interesting point. We found no overlap between our experimentally evolved mutations and naturally occurring *lpxC* mutations, either at the level of nucleotides or codons. However, it is unclear if we should expect to see an overlap for two reasons:

The mutations present in natural isolates likely reflect a combination of beneficial mutations, neutral mutations, and weakly deleterious mutations. The mutations found in our evolved isolates, on the other hand, are all mutations that were beneficial under colistin selection. As such, it is probably not reasonable to expect a strong overlap between the two sets of mutations.The *lpxC* mutations that we observed in our 11 *lpxC* mutated isolates are highly diverse – we found no cases of parallel evolution at the nucleotide level, and only a single example of parallel evolution at the codon level. Given this, our data suggest that a very wide diversity of sites of *lpxC* can interact epistatically with MCR-1 to increase colistin resistance. Again, this high diversity of potential *lpxC* mutations should give a weak association between lab evolved and clinical isolates.

We have added these points in the text (lines 278-304, pages 11-12).